# Symbolic Graph Reasoning Meets Convolutions

**Xiaodan Liang**[1] **, Zhiting hu**[2] **, Hao Zhang**[2] **, Liang Lin**[3] **, Eric P. Xing**[4]
[1] School of Intelligent Systems Engineering, Sun Yat-sen University
[2]Carnegie Mellon University
[3] School of Data and Computer Science, Sun Yat-sen University
[4]Petuum Inc.
`xdliang328@gmail.com, {zhitingh,hao, epxing}@cs.cmu.edu, linliang@ieee.org`

## Abstract

Beyond local convolution networks, we explore how to harness various external
human knowledge for endowing the networks with the capability of semantic
global reasoning. Rather than using separate graphical models (e.g. CRF) or
constraints for modeling broader dependencies, we propose a new Symbolic Graph
Reasoning (SGR) layer, which performs reasoning over a group of symbolic nodes
whose outputs explicitly represent different properties of each semantic in a prior
knowledge graph. To cooperate with local convolutions, each SGR is constituted
by three modules: a) a primal local-to-semantic voting module where the features
of all symbolic nodes are generated by voting from local representations; b) a
graph reasoning module propagates information over knowledge graph to achieve
global semantic coherency; c) a dual semantic-to-local mapping module learns
new associations of the evolved symbolic nodes with local representations, and
accordingly enhances local features. The SGR layer can be injected between
any convolution layers and instantiated with distinct prior graphs. Extensive
experiments show incorporating SGR significantly improves plain ConvNets on
three semantic segmentation tasks and one image classification task. More analyses
show the SGR layer learns shared symbolic representations for domains/datasets
with the different label set given a universal knowledge graph, demonstrating its
superior generalization capability.

## 1 Introduction

Despite significant advances in standard recognition tasks such as image classification [12] and
segmentation [6] achieved by convolution networks, the dominant paradigm lies in the stack of deeper
and complicated local convolutions, and we hope it captures everything about the relationship between
inputs and targets. But such networks compromise the feature interpretability and also lack the global
reasoning capability that is crucial for complicated real-world tasks. Some works [51, 41, 5] thus
formulated graphical models and structure constraints (e.g. CRF [22, 19]) as recurrent works to effect
on final convolution predictions. However, they cannot explicitly enhance feature representations,
leading to the limited generalization capability. The very recent capsule network [39, 14] extends
to learn the sharing of knowledge across locations to find feature clusters, but it can only exploit
implicit and uncontrollable feature hierarchy. As emphasized in [3], visual reasoning over external
knowledge is crucial for human decision-making. The lack of explicitly reasoning over contexts and
high-level semantics would hinder the advances of convolution networks in recognizing objects in a
large concept vocabulary where exploring semantic correlations and constraints plays an important
role. On the other hand, structured knowledge provides rich cues to record human observations and
commonsense using symbolic words (e.g. nouns or predicates). It is thus desirable to bridge symbolic
semantics with learned local feature representations for better graph reasoning.

In this paper, we explore how to incorporate rich commonsense human knowledge [33, 53] into intermediate feature representation learning beyond local convolutions, and further achieve global semantic coherency. The commonsense human knowledge can be formed as various undirected graphs consisting of rich relationships (e.g. semantic hierarchy, spatial/action interactions and attributes, concurrence) among concepts. For example, "Shetland Sheepdog" and "Husky" share one superclass "dog" due to some common characteristics; people wear a hat and play guitar not vice-versa; orange is yellow color. After associating structured knowledge with the visual domain, all these symbolic entities (e.g. dog) can be connected with visual evidence from images, and human can thus integrate visual appearance and commonsense knowledge to help recognize.

We attempt to mimic this reasoning procedure and integrate it into convolution networks, that is, first characterize representations of different symbolic nodes by voting from local features; then perform graph reasoning for enhancing visual evidence of these symbolic nodes via graph propagation to achieve semantic coherency; finally mapping the evolved features of symbolic nodes back into facilitating each local representation. Our work takes an important next step beyond prior approaches in that it directly incorporates the reasoning over external knowledge graph into local feature learning, called as Symbolic Graph Reasoning (SGR) layer. Note that, here we use "Symbolic" to denote nodes with explicit linguistic meaning rather than conventional/hidden graph nodes used in graphical models or graph neural networks [40, 18].

The core of our SGR layer consists of three modules, as illustrated in Figure 1. First, personalized visual evidence of each symbolic node can be produced by voting from all local representations, named as a local-to-semantic voting module. The voting weights stand for the semantic agreement confidence of each local features to a certain node. Second, given a prior knowledge graph, the graph reasoning module is instantiated to propagate information over this graph for evolving visual features of all symbolic nodes. Finally, a dual semantic-to-local module learns appropriate associations between the evolved symbolic nodes and local features to join forces of local and global reasoning. It thus enables the evolved knowledge of a specific symbolic node to only drive the recognition of semantically compatible local features with the help of global reasoning.

The key merits of our SGR layer lie in three aspects: a) local convolutions and global reasoning facilitated with commonsense knowledge can collaborate by learning associations between image-specific observations with prior knowledge graph; b) each local feature is enhanced by its correlated incoming local features whereas in standard local convolutions it is only based on a comparison between its own incoming features and a learned weight vector; c) benefiting from the learned representations of universal symbolic nodes, the learned SGR layer can be easily transferable to other dataset domain with discrepant concept sets. And SGR layer can be plugged between any convolution layers and personalized according to distinct knowledge graphs.

Extensive experiments show superior performance over plain ConvNets by incorporating our SGR layer, especially on recognizing a large concept vocabulary in three semantic segmentation datasets (COCO-Stuff, ADE20K, PASCAL-Context) and image classification dataset (CIFAR100). We further demonstrate its promising generalization capability when transferring SGR layer trained one domain into other domains.

## 2 Related Work

Recent researches that explored the context modeling for convolution networks can be categorized into two streams. One stream exploits networks for the graph-structured data with a family of graph-based CNNs [36, 40] and RNNs [25, 26] or advanced convolution filters [43] to discover more complex feature dependencies. In the context of convolutional networks, the graphical models such as conditional random fields (CRF) [22, 19] can be formulated into a recurrent network by functioning on final predictions of basic convolutions [51, 41, 5]. In contrast, the proposed SGR layer can be treated as a simple feedforward layer that can be injected between any convolution layers and general-purposed for any networks for large-scale and semantic related recognition. Our work differs in that local features are mapped into meaningful symbolic nodes. The global reasoning over locations is directly aligned with external knowledge rather than implicit feature clusters, which is a more effective and interpretable way to introduce structure constraints.

Another stream explored external knowledge bases into facilitating networks. For example, Deng et al. [9] employed a label relation graph to guide network learning while Ordonez et al. [37] learned the

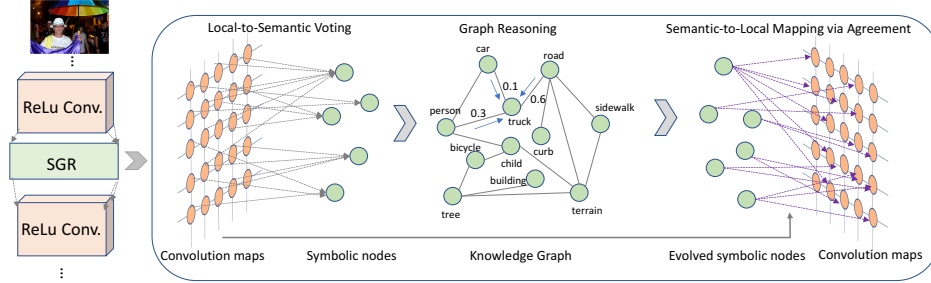

Figure 1: An overview of the proposed SGR layer. Each symbolic node receives votes from all local features via a local-to-semantic voting module (long gray arrows), and its evolved features after graph reasoning are then mapped back to each location via a semantic-to-local mapping module (long purple arrows). For simplicity, we omit more edges and symbolic nodes in the knowledge graph.

mapping of common concepts to entry-level concepts. Some works regularized the output of networks by resorting to complex graphical inference [9], hierarchical loss [38] or word embedding priors [49] on final prediction scores. However, their loss constraints can only function on final prediction layer and indirectly guide visual features to be hierarchy-aware, which is hard to be guaranteed. More recently, Marino et al. [32] used structure prior knowledge to enhance predictions of multi-label classification while our SGR proposes a general neural layer that can be injected into any convolution layers and allows the neural network to leverage semantic constraints derived from various human knowledge. Chen et al. [7] leverage local region-based reasoning and global reasoning to facilitate object detection. In contrast, our SGR layer directly performs reasoning over symbolic nodes and is seamlessly interacted with local convolution layers for better flexibility. Notably, the earliest efforts in reasoning in artificial intelligence date back to symbolic approaches [35] by performing reasoning over abstract symbols with the language of mathematics and logic. After grounding these symbols, statistical learning algorithm [23] is used to extract useful patterns to perform relational reasoning on knowledge bases. An effective reasoning procedure that would be practical enough for advanced tasks should join the force of local visual representation learning and global semantic graph reasoning. Our reasoning layer relates to this line of research by explicitly reasoning over visual evidence of language entities by voting from local representations.

## 3 Symbolic Graph Reasoning

### 3.1 General-purposed Graph Construction

The commonsense knowledge graph is used to depict distinct correlations between entities (e.g. classes, attributes and relationships) in general, which can be any forms. To support the general purposed graph reasoning, the knowledge graph can be formulated as $\mathcal{G} = (\mathcal{N}, \mathcal{E})$, where $\mathcal{N}$ and $\mathcal{E}$ denote the symbol set and edge set, respectively. Here we give three examples: a) class hierarchy graph is constructed by a list of entity classes (e.g. person, motorcyclist) and its graph edges shoulder the responsibility of concept belongings (e.g. "is kind of" or "is part of"). The networks equipped by such hierarchy knowledge can encourage the learning of feature hierarchy by passing the shared representations of parent classes into its child nodes; b) class occurrence graph defines the edges as the occurrence of two classes across images, characterizing the rationality of predictions; c) as a higher-level semantic abstraction, a semantic relationship graph can extend symbolic nodes to include more actions (e.g. "ride", "play"), layouts (e.g. "on top of") and attributes (e.g. color or shape) while graph edges are statistically collected from language descriptions. Incorporating such high-level commonsense knowledge can facilitate networks to prune spurious explanations after knowing the relationship of each entity pair, resulting in good semantic coherency.

Based on this general formula, the graph reasoning is required to be compatible and general enough for soft graph edges (e.g. occurrence probabilities) and hard edges (e.g. belongings), as well as diverse symbolic nodes. Various structure constraints can thus be modeled as edge connections over symbolic nodes, just like human use language tools. Our SGR layer is designed to achieve the general graph reasoning that is applicable for encoding a wide range of knowledge graph forms. As illustrated in Figure 1, it consists of a local-to-semantic voting module, a graph reasoning module and a semantic-to-local mapping module, as presented in following sections.

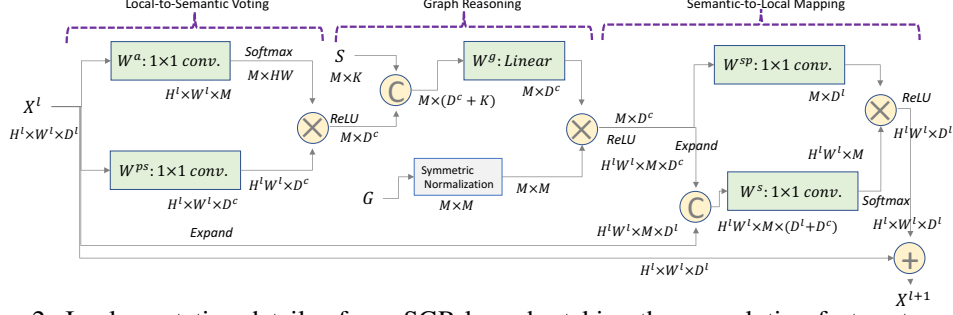

Figure 2: Implementation details of one SGR layer by taking the convolution feature tensors of $H^l \times W^l \times D^l$ as inputs. $\otimes$ denotes matrix multiplication, and $\oplus$ denotes element-wise summation and the circle with C denotes the concatenation. The softmax operation, tensor expansion, ReLU operation are performed when noted. The green boxes denote $1 \times 1$ convolution or linear layer.

## 3.2 Local-to-Semantic Voting Module

Given local feature tensors from convolution layers, our target is to leverage global graph reasoning to enhance local features with external structured knowledge. We thus first summarize the global information encoded in local features into representations of symbolic nodes, that is, local features that are correlated to a specific semantic meaning (e.g. cat) are aggregated to depict the characteristic of its corresponding symbolic node. Formally, we use the feature tensor $X^l \in \mathbf{R}^{H^l \times W^l \times D^l}$ after $l$-th convolution layer as the module inputs, where $H^l$ and $W^l$ are height and weight of feature maps and $D^l$ is the channel number. This module aims to produce visual representations $H^{ps} \in \mathbf{R}^{M \times D^c}$ of all $M = |\mathcal{N}|$ symbolic nodes using $X^l$, where $D^c$ is the desired feature dimension for each node $n$, which is formulated as the function $\phi$:

$$H^{ps} = \phi(A^{ps}, X^l, W^{ps}), \tag{1}$$

where $W^{ps} \in \mathbf{R}^{D^l \times D^c}$ is the trainable transformation matrix for converting each local feature $x_i \in X^l$ into the dimension $D^c$, and $A^{ps} \in \mathbf{R}^{H^l \times W^l \times M}$ denotes the voting weights of all local features to each symbolic node. Specifically, visual features $H_n^{ps} \in H^{ps}$ of each node $n$ are computed by summing up all weighted transformed local features via the voting weight $a_{x_i \to n} \in A^{ps}$ that represents the confidence of assigning local feature $x_i$ to the node $n$. More specifically, the function $\phi$ is computed as:

$$H_n^{ps} = \sum_{x_i} a_{x_i \to n} x_i W^{ps}, \quad a_{x_i \to n} = \frac{\exp(W_n^{a T} x_i)}{\sum_{n \in \mathcal{N}} \exp(W_n^{a T} x_i)}. \tag{2}$$

Here $W^a = \{W_n^a\} \in \mathbf{R}^{D^l \times M}$ is a trainable weight matrix for calculating voting weights. $A^{ps}$ is normalized by using a softmax at each location. In this way, different local features can adaptively vote to representations of distinct symbolic nodes.

## 3.3 Graph Reasoning Module

Based on visual evidence of symbolic nodes, the reasoning guided by structured knowledge is employed to leverage semantic constraints from human commonsense to evolve global representations of symbolic nodes. Here, we incorporate both linguistic embedding of each symbolic node and knowledge connections (i.e. node edges) for performing graph reasoning. Formally, for each symbolic node $n \in \mathcal{N}$, we use the off-the-shelf word vectors [17] as its linguistic embedding, denoted as $\mathcal{S} = \{s_n\}, s_n \in \mathbf{R}^K$. The graph reasoning module performs graph propagation over representations $H^{ps}$ of all symbolic nodes via the matrix multiplication form, resulting in the evolved features $H^g$:

$$H^g = \sigma(A^g B W^g), \tag{3}$$

where $B = [\sigma(H^{ps}), \mathcal{S}] \in \mathbf{R}^{M \times (D^c + K)}$ concatenates features of transformed $H^{ps}$ via the activation function $\sigma(\cdot)$ and the linguistic embedding $\mathcal{S}$. $W^g \in \mathbf{R}^{(D^c+K) \times (D^c)}$ is a trainable weight matrix. The node adjacency weight $a_{n \to n'} \in A^g$ is defined according the edge connections in $(n, n') \in \mathcal{E}$. As discussed in Section 3.1, the edge connections can be soft weights (e.g. 0.8) or hard weight (i.e. {0,1}) according to different knowledge graph resources. The naive multiplication with $A^g$ will

completely change the scale of the feature vectors. Inspired from graph convolutional networks [18], we can normalize $A^g$ such that all rows sum to one to get rid of this problem, i.e. $Q^{-\frac{1}{2}} A^g Q^{-\frac{1}{2}}$, where $Q$ is the diagonal node degree matrix of $A^g$. This symmetric normalization corresponds to taking the average of neighboring node features. This formulation arrives at the new propagation rule:

$$H^g = \sigma(\hat{Q}^{-\frac{1}{2}} \hat{A}^g \hat{Q}^{-\frac{1}{2}} B W^g),\tag{4}$$

where $\hat{A}^g = A^g + I$ is the adjacency matrix of the graph $\mathcal{G}$ with added self-connections for considering its own representation of each node and $I$ is the identity matrix. $\hat{Q}_{ii} = \sum_j \hat{A}^g_{ij}$.

### 3.4 Semantic-to-Local Mapping Module

Finally, the evolved global representations $H^g \in \mathbf{R}^{M \times D^c}$ of symbolic nodes can be used to further boost the capability of each local feature representation. As the feature distributions of each symbolic node have been changed after graph reasoning, a critical question is how to find most appropriate mappings from the representation $h^g \in H^g$ of each symbolic node to all $x_i$. This can be agnostic to learning the compatibility matrix between local features and symbolic nodes. Inspired by message-passing algorithms [11], we compute the mapping weights $a_{h^g \to x_i} \in A^{sp}$ by evaluating the compatibility of each symbolic node $h^g$ with each local feature $x_i$:

$$a_{h^g \to x_i} = \frac{\exp(W^{sT}[h^g, x_i])}{\sum_{x_i} \exp(W^{sT}[h^g, x_i])},\tag{5}$$

where $W^s \in \mathbf{R}^{D^l + D^c}$ is a trainable weight matrix. The compatibility matrix $A^{sp} \in \mathbf{R}^{H \times W \times M}$ is again row-normalized. The evolved features $X^{l+1}$ by graph reasoning, posed as inputs in the $l + 1$ convolution layer can be updated as:

$$X^{l+1} = \sigma(A^{sp} H^g W^{sp}) + X^l,\tag{6}$$

where $W^{sp} \in \mathbf{R}^{D^c \times D^l}$ is the trainable matrix for transforming the dimension of symbolic node representation back into $D^l$, and we use residual connection [12] to further enhance local representations with the original local feature tensor $X^l$. Each local feature is updated by the weighted mappings from each symbolic node that represents different characteristics of semantics.

### 3.5 Symbolic Graph Reasoning Layer

Each symbolic graph reasoning layer is constituted by the stack of a local-to-semantic voting module, a graph reasoning module, and a semantic-to-local mapping module. The SGR layer is instantiated by specific knowledge graph with different numbers of symbolic nodes and distinct node connections. Combining multiple SGR layers with distinct knowledge graphs into convolutional networks can lead to hybrid graph reasoning behaviors. We implement the modules of each SGR via the combination of $1 \times 1$ convolution operations and non-linear functions, detailed as Figure 2. Our SGR is flexible and general enough for injecting it between any local convolutions. Nonetheless, as SGR is designated to incorporate high-level semantic reasoning, using SGR in later convolution layers is more preferable, as demonstrated in our experiments.

## 4 Experiments

As we present the proposed SGR layer as a conventional module suitable for any convolution networks, we thus compare it with on both the pixel-level prediction task (i.e. semantic segmentation) on Coco-Stuff [4], Pascal-Context [34] and ADE20K [52], and image classification task on CIFAR-100 [21]. Extensive ablation studies are conducted on Coco-Stuff dataset [4].

### 4.1 Semantic Segmentation

**Dataset.** We evaluate on three public benchmarks for segmenting over large-scale categories, which pose more realistic challenges than other small segmentation datasets (e.g. PASCAL-VOC) and can better validate the necessity of global symbolic reasoning. Specifically, **Coco-Stuff [4]** contains 10,000 images with dense annotations of 91 thing (e.g. book, clock) and 91 stuff classes (e.g. flower,

| Method | Class acc. | acc. | mean IoU |
|---|---|---|---|
| FCN [31] | 38.5 | 60.4 | 27.2 |
| DeepLabv2 (ResNet-101) [6] | 45.5 | 65.1 | 34.4 |
| DAG RNN + CRF [42] | 42.8 | 63.0 | 31.2 |
| OHE + DC + FCN [15] | 45.8 | 66.6 | 34.3 |
| DSSPN (ResNet-101) [27] | 47.0 | 68.5 | 36.2 |
| SGR (w/o residual) | 47.9 | 68.4 | 38.1 |
| SGR (scene graph) | 49.1 | 69.6 | 38.3 |
| SGR (concurrence graph) | 48.6 | 69.5 | 38.4 |
| SGR (w/o mapping) | 47.3 | 67.9 | 37.2 |
| SGR (ConvBlock4) | 47.6 | 68.3 | 37.5 |
| Our SGR (ResNet-101) | 49.3 | 69.9 | 38.7 |
| Our SGR (ResNet-101 2-layer) | 49.4 | 69.7 | 38.8 |
| Our SGR (ResNet-101 Hybrid) | **49.8** | **70.5** | **39.1** |

Table 1: Comparison on Coco-Stuff test set (%). All our models are based on ResNet-101.

| Method | mean IoU (%) |
|---|---|
| FCN [31] | 37.8 |
| CRF-RNN [51] | 39.3 |
| ParseNet [30] | 40.4 |
| BoxSup [8] | 40.5 |
| HO CRF [1] | 41.3 |
| Piecewise [29] | 43.3 |
| VeryDeep [44] | 44.5 |
| DeepLab-v2 (ResNet-101) [6] | 45.7 |
| RefineNet (Res152) [28] | 47.3 |
| Our SGR (ResNet-101) | 50.8 |
| Our SGR (Transfer convs) | 51.3 |
| Our SGR (Transfer SGR) | **52.5** |

Table 2: Comparison on PASCAL-Context test set(%).

wood), including 9,000 for training and 1,000 for testing. **ADE20k [52]** consists of 20,210 images for training and 2,000 for validation, annotated with 150 semantic concepts (e.g. painting, lamp). **PASCAL-Context [34]** includes 4,998 images for training and 5105 for testing, annotated with 59 object categories and one background. We use standard evaluation metrics of pixel accuracy (pixAcc) and mean Intersection of Union (mIoU).

**Implementation.** We conduct all experiments using Pytorch, 2 GTX TITAN X 12GB cards on a single server. We use the Imagenet-pretrained ResNet-101 [12] as basic ConvNet following the procedure of [6], employ *output stride = 8* and incorporate the SGR layer into it. The detailed implementation of one SGR layer is in Figure 2. Our final SGR model first employs the Atrous Spatial Pyramid Pooling (ASSP) [6] modules with pyramids of {6,12,18,24} to reduce 2,048-d features from final ResBlock of ResNet-101 into 256-d features. Upon this, we stack one SGR layer to enhance local features and then a final $1 \times 1$ convolution layer to produce final pixel-wise predictions. $D^l$ and $D^c$ for feature dimensions in both local-to-semantic voting module and graph reasoning module are thus set as 256, and we use ReLU activation function for $\sigma(\cdot)$. Word embeddings from fastText [17] are used to represent each class, which extracts sub-word information and generalizes well to out-of-vocabulary words, resulting in a $K = 100$-d vector for each node.

We use a universal concept hierarchy for all datasets. Following [27], starting from the label hierarchy of COCO-Stuff [4] that includes 182 concepts and 27 super-classes, we manually merge concepts from the rest two dataset together by using WordTree as [27]. It results in 340 concepts in the final concept graph. Thus, this concept graph makes the symbolic graph reasoning layer can be identical across all three datasets and its weights can be easily shared to each other dataset. We fix the moving means and variations in batch normalization of ResNet-101 during finetuning. We adopt the standard SGD optimization. Inspired by [6], we use the "poly" learning rate policy, set the base learning rate to 2.5e-3 for newly initialized layers and 2.5e-4 for pretrained layers. We train 64 epochs for Coco-Stuff and PASCAL-Context, and 120 epochs for ADE20K dataset. For data augmentation, we adopt random flipping, random cropping and random resize between 0.5 and 2 for all datasets. Due to the GPU memory limitation, the batch size is used as 6. The input crop size is set as $513 \times 513$.

### 4.1.1 Comparison with the state-of-the-arts

Table 1, 2, 3 report the comparisons with recent state-of-the-art methods on Coco-Stuff, Pascal-Context and ADE20K dataset, respectively. Incorporating our SGR layer significantly outperforms existing methods on all three datasets, demonstrating its effectiveness of performing explicit graph reasoning beyond local convolutions for large-scale pixel-level recognition. Figure 3 shows the qualitative comparison with the baseline "Deeplabv2 [6]". Our SGR obtains better segmentation performance, especially for some rare classes (e.g. umbrella, teddy bear), benefiting from the joint reasoning with frequent concepts over the concept hierarchy graph. Particularly, applying the techniques of incorporating high-level semantic constraints designed for classification task into pixel-wise recognition is not trivial since associating prior knowledge with dense pixels itself is difficult. The prior works [38, 10, 49] also attempt to implicitly facilitate the network learning with the hierarchical classification objective. The very recent DSSPN [27] directly designs a network layer for each parent concept. However, this method is hard to scale up for large-scale concept set and

| Method | mean IoU | pixel acc. |
|---|---|---|
| FCN [31] | 29.39 | 71.32 |
| SegNet [2] | 21.64 | 71.00 |
| DilatedNet [47] | 32.31 | 73.55 |
| CascadeNet [52] | 34.90 | 74.52 |
| ResNet-101, 2 conv [45] | 39.40 | 79.07 |
| PSPNet (ResNet-101)DA_AL [50] | 41.96 | 80.64 |
| Conditional Softmax [38] | 31.27 | 72.23 |
| Word2Vec [10] | 29.18 | 71.31 |
| Joint-Cosine [49] | 31.52 | 73.15 |
| DeepLabv2 (ResNet-101) [6] | 38.97 | 79.01 |
| DSSPN (ResNet-101) [27] | 42.03 | 81.21 |
| Our SGR (ResNet-101) | **44.32** | **81.43** |

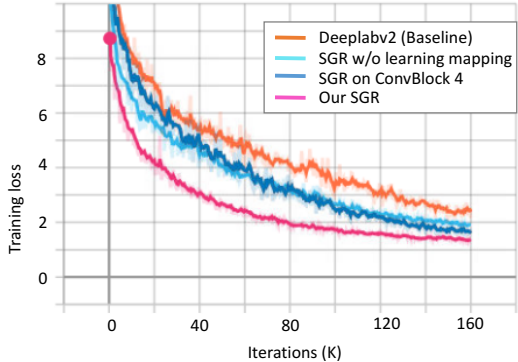

Table 3: Comparison on the ADE20K val set [52] (%). "Conditional Softmax [38]", "Word2Vec [10]" and "Joint-Cosine [49]" use VGG as backbone. We use "DeepLabv2 (ResNet-101) [6]" as baseline.

Table 4: Curves of the training losses on Coco-Stuff for the Deeplabv2 (Baseline) [6] and our three variants. Following [6], the loss is the summations of losses for inputs of three scales (i.e. 1, 0.75, 0.5).

results in redundant predictions for pixels that unlikely belongs to a specific concept. Unlike prior methods, the proposed SGR layer can achieve better results by only adding one reasoning layer while preserving both good computation and memory efficiency.

### 4.1.2 Ablation studies

**Which ConvBlock to add SGR layer?** Table 1 and Table 4 compare the variants of adding a single SGR layer into different stages of ResNet-101. "SGR ConvBlock4" means the SGR layer is added to right before the last residual block of res4 while all other variants add SGR layer before the last residual block of res5 (final residual block). The performance of "SGR ConvBlock4" is worse than "Our SGR (ResNet-101)" while using SGR layer for both res4 and res5 ("Our SGR (ResNet-101 2-layer)") can slightly improve the results. Note that in order to use pretrained weights from ResNet-101, "Our SGR (ResNet-101 2-layer)" directly fuses the prediction results from two SGR layers after res4 and res5 via the summation to get the final prediction. One possible explanation for this observation is that the final res5 can encode more semantically abstracted features, which is more suitable for conducting symbolic graph reasoning. Furthermore, we find removing residual connection in Eqn. 6 would decrease the final performance but is still better than other baselines, by comparing "SGR (w/o residual)" with our full SGR. The reason is that the SGR layer induces more smoothing local features enhanced by global reasoning and thus may degrade some discriminative capability in boundaries.

**The effect of semantic-to-local mapping.** Note that our SGR learns distinct voting weights and mapping weights in the local-to-semantic modules and semantic-to-local module, respectively. The advantages of reevaluating mapping weights can be seen by comparing "Our SGR (ResNet-101)" with "SGR (w/o mapping)" in both testing performance and training convergence in Table 1 and Table 4. This justifies that estimating new semantic-to-local mapping weights can make the reasoning process better accommodate with the evolved feature distributions after graph reasoning, otherwise the evolved symbolic nodes will be misaligned with local features.

**Different prior knowledge graphs.** As discussed in Section 3.1, our SGR layer is general for any forms of knowledge graphs with either soft or hard edge weights. We thus evaluate results of leveraging distinct knowledge graphs in Table 1. First, class concurrence graph is often used to represent the frequency of any two concepts appearing in one image, which depicts inter-class rationality in a statistic view. We calculate the class concurrence graph from all training images on Coco-Stuff and feed it as the input of SGR layer, as "SGR (concurrence graph)". We can see that incorporating a concurrence-driven SGR layer can also boost the segmentation performance, but is slightly inferior to that with concept hierarchy. Second, we also sequentially stack one SGR layer with hierarchy graph and one layer with concurrence graph, leading to a hybrid version as "Our SGR (ResNet-101 Hybrid)". This variant achieves the best performance among all models, verifying the benefits of boosting semantic reasoning capability with the mixtures of knowledge constraints. Finally, we further explore a rich scene graph that includes concepts, attributes and relationships for encoding higher-level semantics, as "SGR (scene graph)" variant. Following [24], the scene graph

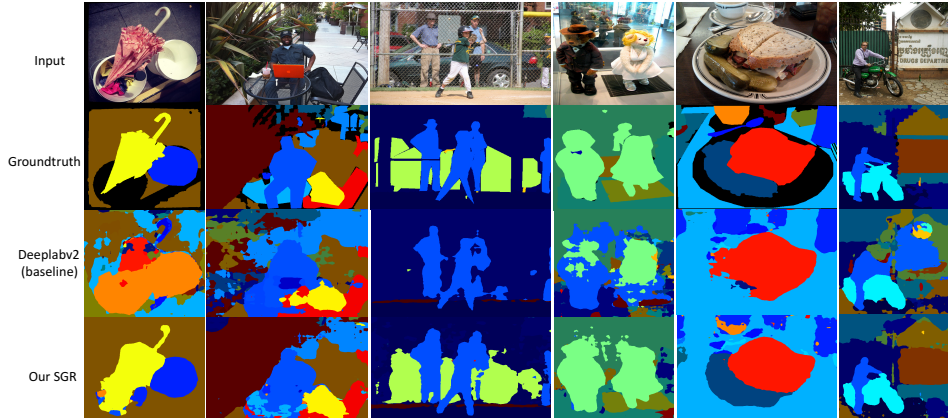

Figure 3: Qualitative comparison results on Coco-stuff dataset.

| Method | ResNet [13] | Wide [48] | ResNeXt-29 [46] | DenseNet [16] | DenseNet-100 [16] (baseline) | SGR | SGR 2-layer |
|---|---|---|---|---|---|---|---|
| Depth | 1001 | 28 | 29 | 190 | 100 | 100+1* | 100+2* |
| Params | 16.1M | 36.5M | 68.1M | 25.6M | 7.0M | 7.5M | 8.1M |
| Error | 22.71 | 20.50 | 17.31 | 17.18 | 22.19 | 17.68 | 17.29 |

Table 5: Comparison of model depth, number of parameters (M), test errors (%) on CIFAR-100. "SGR" and "SGR 2-layer" indicate the results of appending one or two SGR layer on the final denseblock of the baseline network (DenseNet-100), respectively.

is constructed from the Visual Genome [20]. For simplicity, we only select the object categories, attributes, and predicates, which appear at least 30 times and are associated with our targeted 182 concepts in Coco-Stuff. It leads to an undirected graph with 312 object nodes, 160 attribute nodes, and 68 predicate nodes. "SGR (scene graph)" is slightly worse than "Our SGR (ResNet-101)" but better than "SGR (concurrence graph)". Observed from all these studies, we thus use the concept hierarchy graph for all rest experiments by balancing the efficiency and effectiveness.

**Transferring SGR learned from one domain to other domains.** Our SGR layer naturally learns to encode explicit semantic meanings for general symbolic nodes after voting from local features, whose weights can be easily transferred from one domain into other domains only if these domains share one prior graph. Due to the usage of a single hierarchy graph for both Coco-Stuff and PASCAL-Context datasets, we can use the SGR model pretrained on Coco-Stuff to initialize the training on PASCAL-Context dataset, as reported in Table 2. "Our SGR (Transfer convs)" denotes only the pretrained weights of residual blocks are used while "Our SGR (Transfer SGR)" is the variant of further using the parameters of SGR layer. We can see that transferring parameters of SGR layer can give more improvements than that of solely transferring convolution blocks.

### 4.2 Image classification results

We further conduct studies for image classification task on CIFAR-100 [21] consisting of 50K training images and 10K test images in 100 classes. We explore how much SGR will improve the performance of a baseline network, DenseNet-100 [16]. We append SGR layers on the final dense block which produces 342 feature maps with $8 \times 8$ size. We first use a $1 \times 1$ convolution layer to reduce 342-d feature into 128-d, and then sequentially employ one SGR layer, global average pooling and a linear layer to produce final classification. The concept hierarchy graph with 148 symbolic nodes is generated by mapping 100 classes into WordTree, similar to the strategy used in segmentation experiments, included in Supplementary Material. We set $D^l$ and $D^c$ as 128. During training, we use a mini-batch size of 64 on two GPUs using a cosine learning rate scheduling [16] for 600 epochs. More comparisons in Table 5 demonstrate that our SGR can improve the performance of the baseline network, benefiting from the enhanced features via global reasoning. It achieves comparable results with state-of-the-art methods with considerable less model complexity.

## 5 Conclusion

To endow the local convolution networks with the capability of global graph reasoning, we introduce a Symbolic Graph Reasoning (SGR) layer, which harnesses external human knowledge to enhance local feature representation. The proposed SGR layer is general, light-weight and compatible with existing convolution networks, consisting of a local-to-semantic voting module, a graph reasoning module, and a semantic-to-local mapping module. Extensive experiments on both three public benchmarks on semantic segmentation and one image classification dataset demonstrated its superior performance. We hope the design of our SGR can help boost the research of investigating global reasoning property of convolution networks and be beneficial for various applications in the community.

## Acknowledgements

This work was supported in part by the National Key Research and Development Program of China under Grant No. 2018YFC0830103, in part by National High Level Talents Special Support Plan (Ten Thousand Talents Program), and in part by National Natural Science Foundation of China (NSFC) under Grant No. 61622214, and 61836012.

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
