[Reviews · NeurIPS 2018]

Reviewer 1



This paper studies how to inject external human knowledge to neural networks. It proposes a new Symbolic Graph Reasoning (SGR) layer. SGR layer bridges convolution layers and prior graphs. A SGR layer has three components: local-to-Semantic voting, graph reasoning, and semantic-to-local mapping. The proposed method shows improvement in segmentation and classification across multiple datasets: COCO-stuff, ADE20k, and PASCAL-Context and CIFAR-100. Authors proposed a new layer called (SGR). SGR layer maps convolution feature space to semantic space and propagates (graph reasoning) in semantic space. The evolved features are brought back to convolution feature spaces by semantic-to-local mapping. The authors suggest symmetric normalization for the stability which is a well-known trick in graph analysis literature. Due to the simplicity of mapping between convolution feature space and semantic space, the graph reasoning is not effective for lower convolution layers as authors observed. The authors may want to plug-in the graph reasoning with more complicated mapping in some applications where lower level convolution features are important. More discussion on mappings will be useful. The great feature of this work is that knowledge graphs can change the convolution features. If possible, a qualitative comparison between original conv feature maps and evolved feature maps will give more intuition. Most graph reasoning is quite sensitive to how to build the graph. It was not discussed. The knowledge graphs can be viewed as a transition matrix. Given a knowledge graph, multiple multiplications of transition matrix offline will yield different interpolation effects on the discrete topological space. It will be interesting to see different behaviors of the proposed layer. In Visual Genome dataset, the relationships between objects are not symmetric. Due to the symmetric normalization in the SGR layer, this knowledge can be incorporated naturally. The main difficulty to handle asymmetric (directed) graph is not properly discussed. From the classical graph analysis literature, lots of tricks can be used to keep ergodicity of graphs and stability of propagation. Extension to asymmetric graphs will expand the operation range of the proposed method. Lastly, due to the conv1x1, semantic mapping was done pixel-by-pixel. Region/group-wise semantic mapping can make the proposed method more efficient using larger convolution kernel and deconvolution layer. Current construction increases too much overhead. For example, Additional computational cost. Adding one SGR layer increases 0.5 M parameters in 7.0M network (DenseNet-100). PSPNet on ADE20K, the performance 44.94 and 81.69 which are better than the best performance of the proposed method in Table 3. Further, DSSPN (ResNet-101) [27], the numbers are different from the original paper. Did you run the baseline again? Please clarify this. Overall, the SGR layer is a simple and interesting idea. The empirical results look promising. Even if the proposed method is not very efficient, this will allow incorporating human knowledge in CNNs. I forward to more applications and extensions of this work.

Reviewer 2



This paper combines local convolutional neural networks with symbolic graph reasoning, which consists of three different layers including local-to-semantic attention layer, graph propagation layer, semantic to local mapper layer. Experimental results prove the effectiveness of the proposed approach on the task of semantic segmentation and image classification. Strength: -The paper is very well written and easy to follow -The proposed method seems to be very effective Weakness: - the novelty of the proposed method seems to be straightforward and marginal

Reviewer 3



This paper is about a Symbolic Graph Reasoning (SGR) layer which can be injected between convolutional layers. The SGR layer employs a knowledge graph that allows the neural network to leverage semantic constraints derived from human knowledge. The SGR is structured in 3 modules: 1) local-to-semantic voting module which maps local features to semantic nodes (i.e. the nodes of the knowledge graph) and outputs a D^c dimensional feature vector for each semantic node in the graph, 2) a graph reasoning module in which the semantic node features are concatenated with word-vector features (derived from linguistic embeddings) projected to lower dimensionality linear layer. Finally, each semantic node representation is replaced by the average of the node representations of its neighbors (using the dot product with the row-normalized adjacency matrix of the knowledge graph). 3) the semantic-to-local mapping module reconstructs the local features starting from the global representation of the semantic nodes in produced in the previous module. The paper include experiments with the SGR module on the semantic segmentation datasets such as Coco-Stuff, ADE20K and PASCAL-Context and on the classification dataset CIFAR-100. The experimental evaluation is quite convincing and I find impressive that the performance of SGR/SGR 2-layer can get very close to the one of DenseNet using 7.5M/8.1M of parameters instead of 25.6M (see Table 5). With respect to the novelty, the closest work is the one on Dynamic-structured semantic propagation networks [27] The author claim that "Our work takes an important next step beyond prior approaches in that it directly incorporates the reasoning over external knowledge graph into local feature learning, called as Symbolic Graph Reasoning (SGR) layer." However, also [27] can incorporate a graph and yields results which are a bit lower on Coco-stuff. Strengths: + The paper is well written and easy to read. + The performance of the method is impressive and does not seem to be tailored to the specific tasks addressed. Weaknesses: - It not clear what is novel compared to [27], which also incorporate a graph and yields similar results which are a bit lower on Coco-stuff Questions: Q1) Since [27] is missing in Table 2. Is there a reason because [27] can not be run on Pascal-Context? Q2) In [27] we can see experiments on the Cityscape and the Mapillary datasets would SGR work on those datasets? Q3) "The very recent DSSPN [27] directly designs a network layer for each parent concept. However, this method is hard to scale up for large-scale concept set and results in redundant predictions for pixels that unlikely belongs to a specific concept." Q3.1 Could you provide evidence of a large-scale concept set in which your method surpasses [27]? Q3.2 Is there a relationship between the size of the concept set and the number object classes that we want to segment? Q4) Could you clarify what is meant with the term "reasoning" in SGR? In particular, could we interpret the dot product of the row-normalized adjacency matrix with the semantic node features as a forward chaining step in an inference engine? Minor Issues: - "people wear a hat and play guitar not vice-versa;" why not? - "orange is yellow color." orange is orange not yellow, unless I am missing something. - "Comparison with the state-of-the-arts" -> "Comparison with the state of the art" —————— My concerns are all well addressed in the rebuttal. My opinion of the paper is increased.

Reviewer 4



This paper describes a method to integrate knowledge graph in a deep neural network, which is usually learned end to end by data. To achieve that, a module named "symbolic graph reasoning" layer is used to functionally merge the knowledge graph (node embeddings and adjacency) into the hidden space, to help the subsequent tasks. A good amount of experiments are done on several datasets, and the results are on a par with state-of-art results. The paper is well written. Motivations and methods are laid out nicely. Experiments with the methods are at the same level of the state-of-the-art results for the used dataset. In analaysis, variants of the method was done to isolate the effect of some of the components. Several comments: 1. The coco-stuff results in [27] has a slightly higher number (50.3). Should it be cited as well? 2. The paper: "The More You Know: Using Knowledge Graphs for Image Classification. Kenneth Marino, Ruslan Salakhutdinov, Abhinav Gupta" seems to be a more relevant method to compare to.